# Functional Mechanisms Underlying the Antimicrobial Activity of the *Oryza sativa* Trx-like Protein

**DOI:** 10.3390/ijms20061413

**Published:** 2019-03-20

**Authors:** Seong-Cheol Park, Il Ryong Kim, Jung Eun Hwang, Jin-Young Kim, Young Jun Jung, Wonkyun Choi, Yongjae Lee, Mi-Kyeong Jang, Jung Ro Lee

**Affiliations:** 1Department of Polymer Science and Engineering, Sunchon National University, Suncheon 57922, Korea; schpark9@gnu.ac.kr (S.-C.P.); jyfrog@hanmail.net (J.-Y.K.); 2National Institute of Ecology, 1210 Geumgang-ro, Maseo-myeon, Seocheon-gun 33657, Korea; kir6060@nie.re.kr (I.R.K.); jehwang@nie.re.kr (J.E.H.); jun5763@nie.re.kr (Y.J.J.); wonkyun@nie.re.kr (W.C.); 3Division of Applied Life Science (BK21+ Program) and PMBBRC, Gyeongsang National University, Jinju 52828, Korea; 4Department of Nutrition and Food Science, Texas A&M University, College Station, TX 77843, USA; yongjaelee@tamu.edu; 5The Research Institute for Sanitation and Environment of Coastal Areas, Sunchon National University, Suncheon 57922, Korea

**Keywords:** antifungal activity, plant defense, reactive oxygen species, thioredoxin, tetratricopeptide repeat domain

## Abstract

Plants are constantly subjected to a variety of environmental stresses and have evolved regulatory responses to overcome unfavorable conditions that might reduce or adversely change a plant’s growth or development. Among these, the regulated production of reactive oxygen species (ROS) as a signaling molecule occurs during plant development and pathogen defense. This study demonstrates the possible antifungal activity of *Oryza sativa* Tetratricopeptide Domain-containing thioredoxin (OsTDX) protein against various fungal pathogens. The transcription of OsTDX was induced by various environmental stresses known to elicit the generation of ROS in plant cells. OsTDX protein showed potent antifungal activity, with minimum inhibitory concentrations (MICs) against yeast and filamentous fungi ranging between 1.56 and 6.25 and 50 and 100 µg/mL, respectively. The uptake of SYTOX-Green into fungal cells and efflux of calcein from artificial fungus-like liposomes suggest that its killing mechanism involves membrane permeabilization and damage. In addition, irregular blebs and holes apparent on the surfaces of OsTDX-treated fungal cells indicate the membranolytic action of this protein. Our results suggest that the OsTDX protein represents a potentially useful lead for the development of pathogen-resistant plants.

## 1. Introduction

Crops are exposed to adverse environmental conditions during their growth and survival. In order to cope with diverse biotic or abiotic stress such as pathogen attack, low temperatures, heat stress, and hormones, plants have evolved various defense systems and signaling pathways. Reactive oxygen species (ROS), such as hydrogen peroxide (H_2_O_2_), superoxide radical(O_2_^−^), hydroxyl radical(·OH), and singlet oxygen(^1^O_2_), are signaling molecules involved in various processes including biotic and abiotic stress responses, pathogen defense, and plant development [1,2,3]. To regulate ROS production, plants have evolved networks of various stress-responsive genes and proteins [4]. Some stress-responsive proteins, such as thioredoxin (Trx), catalase (Cat), ascorbate peroxidase (Apx), and peroxiredoxin (Prx), are well known ROS-regulatory proteins. 

Plants are constantly exposed to diverse biotic and abiotic stresses, and have evolved defense mechanisms to overcome such stresses. Previous studies have shown interactions or cross-talk between the signaling pathways involved in the response to biotic and abiotic stresses [5,6,7,8,9]. In these signaling pathways, ROS plays key roles as a trigger of stress responses in plants [10]. Some studies show that salt and heat stress are related to pathogen defense responses [11,12,13]. Ionizing radiations (IRs), which comprise energetic subatomic particles, ions, or high-energy electromagnetic waves, penetrate living tissue of organisms and cause biological damage via the generation of ROS in cells [14,15,16]. Under natural conditions, plants are exposed to natural IR and non-IR. An understanding of the plant response to a variety of environmental stresses including salt, heat, and IR should facilitate the improvement of plant immunity under environmental oxidative stress.

In this study, we investigated the transcription levels of rice genes that encode ROS regulators in response to environmental stresses. Interestingly, a rice Trx-like protein was induced by several abiotic stresses such as heat, salinity, and IR exposure. The protein was found to be composed of a TRX motif and a TPR domain consisting of a 34-amino-acid sequence. TRX proteins play a significant role in the maintenance of cellular redox homeostasis [17]. In particular, the inhibitory activities of *Arabidopsis thaliana* Trx-h3 and Trx-h5 against pathogenic attack are well known [18,19]. The TPR domain is involved in protein-protein interaction and formation of multicomplex proteins with characteristic chaperone activities [20]. In this study, we characterized the natural antimicrobial activity of a rice Trx-like protein and examined its potential utility as an antimicrobial agent.

## 2. Results and Discussion

### 2.1. Isolation and Functional Characterization of Oryza sativa Tetratricopeptide Domain-Containing Thioredoxin (OsTDX) Protein 

Previous studies have shown that the production of plant redox proteins is induced by biotic and abiotic stresses, and that these proteins are functionally related to defense responses. To investigate the effect of various abiotic stresses on the expression of rice redox genes, rice seed or seedlings were exposed to heat, cold, salt, and IR. Then, quantitative reverse transcription PCR (qRT-PCR) was used to assess the expression level of rice redox genes (Figure 1 and Appendix A). Interestingly, the transcription of a rice Trx-like protein OsTDX (accession no. XP 015612619) was significantly induced in response to heat and salt stress, and slightly changed by two types of IR; however, *OsTDX* gene expression was reduced in response to cold stress (Figure 1). In contrast, transcript levels of other two redox genes did not significantly change in diverse external stresses except for heat condition (Appendix A). Plant Trx proteins have been identified as pathogen- or stress-regulating molecules [11,12,13,18,19,21]. Therefore, we investigated whether OsTDX plays a key role as an antimicrobial protein in plant cells.

To determine antimicrobial activity of OsTDX, we expressed recombinant OsTDX protein in *Escherichia coli*. The full-length cDNA of *OsTDX* was cloned into the pGEX vector and expressed in *E. coli* BL21(DE3). To confirm the purity of isolated recombinant protein, 12% SDS-PAGE analysis was conducted, and the protein identity was confirmed by MALDI-TOF analysis using a major and three faint bands (Figure 2A,C and Appendix A). Peptide sequences of all protein bands on a SDS-PAGE gel exactly matched the partial fragments of *O. sativa* TDX protein (Figure 2A,C and Appendix A). There was a slight difference between its predicted MW (~35 kDa) and the location on the gel (Figure 2A). Native-PAGE, which was performed to determine the native structures of the protein, revealed that OsTDX formed oligomeric complex (Figure 2B). Size-exclusion chromatography (SEC) of the OsTDX protein revealed a major peak that corresponds to a band observed by native-PAGE (Figure 2D). Standard bars of molecular weight ranging from 17 kDa to 670 kDa confirmed oligomeric form of recombinant OsTDX, see Figure 2B. In addition, the forms of OsTDX proteins fractionated on SEC (a, b, and c fractions indicated in Figure 2D) were observed by negative staining on transmission electron microscope (TEM), resulted in irregular high oligomer, regular oligomer, and dimer (a, b, and c in Figure 2E, respectively). Results of these physicochemical analyses indicate that pure recombinant OsTDX was extracted, and that this protein comprises oligomeric structures under native conditions.

### 2.2. Antifungal and Cytotoxic Effects of Recombinant OsTDX

To evaluate the antifungal activity of the recombinant OsTDX protein, we performed a growth inhibitory assay against various fungal pathogens including *Colletotrichum gloeosporioides*, *Fusarium graminearum*, *Fusarium oxysporum*, *Fusarium solani*, *Gibberella zeae*, *Trichosporon beigelii*, *Candida albicans*, *Candida krusei*, and *Candida parapsilosis*. As shown in Table 1, OsTDX inhibited the growth of all tested filamentous and yeast fungal strains. The minimum inhibitory concentrations (MICs) of OsTDX ranged from 50 to 100 µg/mL in filamentous fungi and from 1.56 to 6.25 µg/mL in yeast. In particular, OsTDX was found to exert stronger antifungal activity than melittin, a cytotoxic peptide used as a control, in yeast. Figure 3A–C show the dose-dependent antifungal activities of OsTDX protein against *C. albicans*, *C. krusei*, and *C. gloeosporioides*, respectively. *C. albicans* and *C. krusei* colonies were not formed on agar medium that was further cultivated after 24-h incubation of fungal cells in the presence of OsTDX protein, even at a concentration of 3.13 µg/mL (Figure 3A,B). At a concentration of 100 µg/mL, OsTDX completely inhibited the growth of *C. gloeosporioides* cells (Figure 3C-c). We assumed that the differences between the antifungal activities of OsTDX protein in filamentous and yeast fungi may be attributable to their specific cell wall constituents such as mannan, β-1,6-glucan, β-1,3-glucan, and chitin layers, or cell membrane components such as phospholipids and sterol.

To evaluate the toxic effect of OsTDX protein in mammalian cells, hemolysis assay was performed with rat erythrocytes. Melittin, which is a well-known cytotoxic peptide, induced 86.4% hemolysis at 1.56 µg/mL, while OsTDX protein and bovine serum albumin (BSA) did not elicit hemoglobin release at 400 µg/mL (Figure 3D). These results suggest that the OsTDX protein is a nontoxic antifungal material with potential use in various applications, as it did not show cytotoxic effects at concentrations higher than its antifungal minimum inhibitory concentrations (MICs).

### 2.3. Molecular Mechanism of OsTDX in Fungal Cells

To investigate the cellular distribution of OsTDX in fungal cells, *C. albicans* cells were incubated with fluorescein amidite (FAM)-labeled melittin and OsTDX, and fluorescence was observed by confocal laser scanning microscopy. As shown in Figure 4A, both OsTDX and melittin showed remarkable levels of accumulation in the cell walls and membranes of *C. albicans*. These observations suggest that the growth-inhibitory activity of OsTDX protein against fungal cells may result from the interaction between OsTDX and the fungal membrane.

To confirm the above results, a SYTOX-Green uptake assay was performed using a fluorescence spectrophotometer and flow cytometry. SYTOX-Green penetrates the cell by inducing cell membrane damage, and its green fluorescence increases by binding to cytoplasmic nucleic acids. Melittin showed rapid uptake of vital dyes during the first 20 min; this was followed by their gradual uptake in the presence of OsTDX, but at a rate that was higher than that of melittin after 20 min under the same conditions. After 30 min, the uptake of both OsTDX and melittin reached stationary phase (Figure 4B). The SYTOX-Green assay with various concentrations of OsTDX and melittin was performed to investigate a dose-dependent membrane damage (Figure 4C). Melittin was well known to disrupt fungal membrane via pore formation mechanism [22]. Although we did not observe the membrane pore directly in TEM as melittin, we suggest that OsTDX protein exhibits a potent antifungal activity via destabilizing cell membrane or membranolysis. 

### 2.4. Effects of OsTDX on Membrane Permeability

To further examine the effect of OsTDX on cell membranes, the membrane-permeabilizing abilities of the OsTDX protein were investigated by measuring the calcein released from artificial liposomes, which comprised phosphatidylcholine (PC):phosphatidylethanolamine (PE): phosphatidylinositol (PI):ergosterol (5:4:1:2, *w*/*w*/*w*/*w*) as a fungal membrane and PC:cholesterol (Ch):sphingomyelin (SM) (1:1:1, *w*/*w*/*w*) as a mammalian membrane. As shown in Figure 5A, in the presence of melittin and OsTDX, fungal liposomes showed a significant release of calcein in a dose-dependent manner. However, OsTDX and BSA did not induce the release of calcein from mammalian liposomes, whereas melittin elicited remarkable membrane damage at the lowest tested concentration (Figure 5B). Results from the investigation of membrane-permeabilizing activities were consistent with those demonstrating antifungal and cytotoxic effects. 

### 2.5. Morphological Alterations Caused by OsTDX in Fungal Cells

In order to investigate antifungal activity, the morphology of protein-treated mold and yeast cells was observed under SEM. After 4-h incubation with OsTDX or melittin at each MIC, *C. gloeosporioides* or *C. albicans* cells showed a crumpled appearance on the whole, with irregular-sized blebs and holes in the cell surface; in contrast, smooth surfaces were observed in the absence of proteins (Figure 5C,D). Previous reports based on physiological and morphological data for artificial liposomes suggest that melittin forms toroidal pores in the surfaces of fungal cells [22,23]. Fungal cell walls are generally composed of chitin, β-1,3-glucan, β-1,6-glucan, mannan, and proteins [24]. In particular, we assumed that abnormal bubbles are produced in cell walls as a result of osmotic pressure, which induces the efflux of cytosolic materials due to the disruption of membranes and formation of holes by OsTDX or melittin. Therefore, the antifungal mechanism of OsTDX protein is suggested to be membranolytic.

## 3. Materials and Methods 

### 3.1. Materials

Bovine serum albumin (BSA), Ch, ergosterol, and SM were purchased from Sigma-Aldrich Co. (St. Louis, MO, USA). PC, PE, and PI were bought from Avanti^®^ Polar Lipids, Inc. (Alabaster, AL, USA). MitoSOX Red, 2′,7′-dichlorofluorescein diacetate (DCFH-DA), and SYTOX-Green were obtained from Molecular Probes Inc., (Eugene, OR, USA). FAM *N*-hydroxysuccinimide (NHS) ester was obtained from BioActs (Incheon, Korea). All other reagents were of analytical grade.

Rice seeds (*Oryza sativa* cv. Ilpoom) were used to determine the response of rice to abiotic stress. Rice seeds were germinated and grown in half-strength MS medium (Duchefa Biochemie, Haarlem, Netherlands) containing 3% sucrose (Sigma, St. Louis, MO, USA) and 0.7% phytoagar at 24 °C with photoperiod of 16 h light/8 h dark and 70% humidity. After 10 days, seedlings were treated with heat, cold, and salt. For the cold and heat stress treatments, the seedlings were placed in a temperature-controlled chamber at 4 °C or 45 °C for 48 h, respectively. For the salt stress treatment, the seedlings were transferred to new medium containing 200 mM NaCl for 48 h. In order to examine the plant response to ionizing radiation, seeds were exposed to GA (gamma ray), IB (ion beam), and CR (cosmic ray). For the GA treatment, rice seeds were irradiated with 200 Gy of gamma radiation generated by a gamma irradiator (^60^Co, ca. 150 TBq of capacity, ACEL, MDS Nordion, ON, Canada) for 24 h at the Korea Atomic Energy Research Institute. The ion beam treatment consisted of irradiation with 220 MeV carbon ions (LET 107 keV/µm), at a dose of 40 Gy, generated by an AVF cyclotron (Japan Atomic Energy Agency, Takasaki, Japan). Exposure to CR was achieved by placing the samples on the ‘Shijian-8′, an unmanned breeding spacecraft, for 15 days. The irradiated seeds were grown on half-strength MS medium containing 3% sucrose and 0.7% phytoagar at 24 °C with photoperiod of 16 h light/8 h dark and 70% humidity. To perform gene expression analysis, these seedlings were harvested at 12 days after sowing.

### 3.2. Fungal Cells

*C. gloeosporioides* (KCTC6169), *F. graminearum* (KCTC16656), *F. oxysporum* (KCTC16909), *F.solani* (KCTC6326), *G. zeae* (KCTC6150), *T. beigelii* (KCTC7707), *C. albicans* (KCTC7270), *C. krusei* (CCARM14017), and *C. parapsilosis* (CCARM14016) were obtained from the Korea Collection for Type Cultures (KCTC) and Culture Collection of Antimicrobial Resistant Microbes (CCARM).

### 3.3. RNA Isolation and qRT-PCR

Total RNA was isolated from 12-day-old rice seedlings using TRIzol reagent, as described by the manufacturer (GibcoBRL, Carlsbad, CA, USA). Reverse transcription was performed for 60 min at 42 °C using a Power cDNA Synthesis Kit (Intron Biotech Inc., Sungnam, Korea) with 1 μg oligo(dT)15 primers and 1 μg total RNA as template. The resultant cDNA was used as a template for quantitative RT-PCR. Quantitative RT-PCR was performed using a Bio-Rad CFX qRT–PCR detection system (Bio-Rad Laboratories Inc., Hercules, CA, USA) with iQ™ SYBR^®^ Green supermix (Bio-Rad). The reaction was performed under the following conditions: 95 °C for 2 min, followed by 45 cycles of 95 °C for 30 s and 60 °C for 30 s and 72 °C for 5 min. Each data point is the average of three independent amplifications of the same RNA sample run in the same reaction plate. qRT-PCR analysis was performed using specific primers. DNA was amplified using the following primers. *OsTDX* forward (5′-GCCTGTCTAGGCTTGTGGTT-3′) and *OsTDX* reverse (5′-TCCAGCGATATGCGACACTG-3′); *OsTrx type m* forward (5′-ACGGACGACTCACCAAACAT-3′) and *OsTrx type m* reverse (5′- GCTGCTGACGTACTTGTCGAT-3′); *OsTrx type x* forward (5′-GGATGGGAAAGAGGTACCAGG-3′) and *OsTrx type x* reverse (5′-AGCAACTGTTGAGGTTGACA-3′). The rice Actin gene (*Os03g50885*) was used as an internal reference [25].

### 3.4. Cloning of the OsTDX Gene and Protein Expression in E. coli

The *OsTDX* gene was amplified from a rice leaf cDNA library by PCR. *OsTDX* was cloned into the pGEX-2T vector and the recombinant protein was expressed in *E. coli* strain BL21 (DE3). The glutathione S-transferase (GST)-fused OsTDX protein was incubated with a glutathione-agarose affinity column (Bio-Rad, Hercules, CA, USA), and the GST of the recombinant protein was cleaved by thrombin treatment. Isolated intact OsTDX protein was identified using SDS-PAGE analysis.

### 3.5. Purification and Structural Characterization of OsTDX Protein

Isolated recombinant OsTDX was further purified by fast protein liquid chromatography (FPLC; Bio-Rad). FPLC analysis was performed with a SEC 650 column at a flow rate of 0.5 mL/min at 25 °C with PBS buffer (pH 7.2). The column was calibrated using the gel filtration standards (Bio-Rad) thyroglobulin (670 kDa), γ-globulin (158 kDa), ovalbumin (44 kDa), and myoglobin (17 kDa). The purity of OsTDX protein was determined using 12% SDS-PAGE gel. To identify proteins on SDS-PAGE gel, isolated proteins were digested with trypsin and subjected to peptide mass fingerprint (PMF) using MALDI-TOF analysis (Microflex LRF 20, Bruker Daltonics, Billerica, MA, USA). The mass range was set from 600 to 3000 and calibration was conducted using Trypsin digestion peaks (*m*/*z* 842.5099, 2211.1046). Peaks were picked using Flex Analysis 3.0 (Bruker Daltonics). MASCOT software (http://matrixscience.com) was used for protein identification [26,27]. To characterize a native structure of the recombinant OsTDX protein, native PAGE and transmission electron microscope (TEM) analysis were used. A molecular complex of the protein was displayed on 12% native PAGE gel for 1 h [28,29]. Morphological alteration of OsTDX protein fractionated on SEC was assessed by transmission electron microscope (TEM). For TEM, fractionated proteins were applied to carbon-coated copper grids that had been glow-discharged (Harrick Plasma, Ithaca, NY, USA) in air. Then the grid was negatively stained using 1% uranyl acetate [30]. The grids were examined in a FEI Tecnai 20 TEM operated at 200 kV. Images were recorded using a Gatan CCD camera (1024 × 1024 pixel, Pleasanton, CA, USA).

### 3.6. Antifungal Assay

To determine antifungal activity by microtiter assay, spores of mold fungi from four-day-old cultures grown on potato dextrose broth (PDB, Difco, Sparks, MD, USA) agar plates were collected with 0.08% Triton X-100, and yeast cells were subcultured for overnight in yeast extract peptone dextrose (YPD, Difco, Sparks, MD, USA). Final concentrations of fungal cells were adjusted to 2 × 10^4^ spores/mL in PBS buffer containing appropriate media. The adjusted fungal cells were added to two-fold serially diluted samples in 96-well plates. After 24 h of incubation at 28 °C, hyphal growth and cell growth was monitored microscopically with an inverted light microscope (Carl Zeiss Microscopy, Oberkochen, Germany). The MICs against each fungal species were defined as the lowest concentration of samples that completely inhibited the visible growth. All assays were performed in triplicate [31].

### 3.7. Hemolytic Effect of OsTDX in Rat Red Blood Cells (rRBCs)

Fresh rRBCs from healthy donor rats were centrifuged at 800× *g* and washed with PBS buffer until the supernatant was clear. Two-fold serial dilutions of peptide in PBS were plated and rRBCs (8%, *v*/*v*) were then added to a 96-well plate. The sample was incubated with mild agitation for 1 h at 37 °C and then centrifuged at 800× *g* for 10 min. The absorbance of the supernatant was measured at 414 nm in a microplate reader (SpectraMax M5; Molecular Devices, Sunnyvale, CA, USA); each measurement was made in triplicate, and percentage hemolysis was calculated using the following equation [32,33],
% hemolysis = [(Abs_414 _in the protein solution − Abs_414_ in PBS)/(Abs_414_ in 0.1% Triton-X100 − Abs_414_ in PBS)] × 100

### 3.8. Cellular Distribution of FAM-Labeled OsTDX

In order to label proteins with the fluorescent dye, FAM *N*-Hydroxysuccinimide (NHS) ester, FAM NHS ester solution was added to OsTDX or melittin in PBS (pH 7.4) at a molar ratio of 1:1. After 1 h incubation, the mixture was dialyzed to remove the unreacted fluorescent dye with PBS for 36 h. The cellular localization of FAM-labeled OsTDX or melittin in fungal cells was analyzed by confocal laser scanning microscopy (CLSM). The fungal cell suspensions (10^4^ conidia/mL) were incubated with FAM-labeled OsTDX or melittin at the MICs on a 24-well microtiter plate for 4 h at 28 °C; then, cells were washed with PBS. The washed cells were spotted onto cover slips with mounting solution (50% glycerol and 0.1% *n*-propyl-gallate [34]. The cells were observed under a CLSM (LSM 510 META, Gottingen, Germany). Zeiss LSM imaging software was used for image acquisition and analysis. 

### 3.9. SYTOX-Green Uptake Assay

*Candida albicans* cells were preincubated with 0.5 μM SYTOX-Green for 15 min in the dark were incubated with OsTDX or melittin at MICs or multiple concentrations for 1 h. The cells were then analyzed by spectrofluorometry in a time-dependent manner (excitation wavelength of 485 nm and emission wavelength of 520 nm) and flow cytometry (Attune NxT acoustic focusing cytometer, Thermo Fisher Scientific Co., Waltham, MA, USA).

### 3.10. Calcein Leakage

Calcein-entrapped liposomes were prepared using the previously described freeze-thaw method [35]. In brief, the dried lipids were prehydrated with dye buffer solution (80 mM calcein, 10 mM HEPES, 30 mM NaCl, pH 7.4), vortexed for 1 min, and left for 30 min at 50 °C. The suspension was freeze-thawed in liquid nitrogen for nine cycles to form large unilamellar vesicles (LUVs) and extruded 30 times through polycarbonate filters (two stacked 0.2-μm pore size filters) using an Avanti Mini-Extruder (Avanti Polar Lipids Inc., Alabaster, AL, USA). Calcein-entrapped vesicles were separated from free calcein by gel filtration chromatography on a Sephadex G-50 column, and their concentration was determined via a standard phosphate assay [36]. Entrapped LUVs in suspensions containing 2.5 μM lipids were incubated with various concentrations of the proteins. The fluorescence of the released calcein was assessed using a spectrofluorometer (excitation wavelength, 480 nm; emission wavelength, 520 nm). Complete (100%) release was achieved via the addition of 0.03% (*w*/*v*) Triton X-100. Spontaneous leakage was determined to be negligible at this time scale. The apparent percentage of calcein release was calculated using the following equation [37,38],
Leakage (%) = 100 × (F − F_0_)/(F_t_ − F_0_)
where F = the observed fluorescence in the presence of peptide, F_0_ = the fluorescence of spontaneous leakage (only buffer), and F_t_ = the observed fluorescence after adding of Triton X-100.

### 3.11. Morphological Observation by Scanning Electron Microscopy (SEM)

OsTDX or melittin was incubated with precultivated *C. gloeosporioides* or *C. albicans* cells (1 × 10^6^ cells/mL) at MICs. After 4 h incubation, cells were fixed with 2% glutaraldehyde in 100 mM HEPES buffer (pH 8.0) overnight at 4 °C. The fixed cells were dehydrated in graded ethanol and critical point-dried under CO_2_. The gold-coated samples were observed using a field emission SEM (JSM-7100F, JEOL Ltd., Tokyo, Japan) [39].

## 4. Conclusions

In summary, this study reports the antifungal activity of OsTDX protein against mold and yeast. The antifungal activity of this protein was stronger in yeast than in mold fungi owing to the differences in their cell wall components. OsTDX protein exerted killing action against fungal pathogens via destabilizing and disrupting effects on fungal membranes. The present findings suggest that OsTDX proteins may have potential for the development of pathogen-resistant crop plants or environment friendly agricultural antifungal agents.

## Figures and Tables

**Figure 1 ijms-20-01413-f001:**
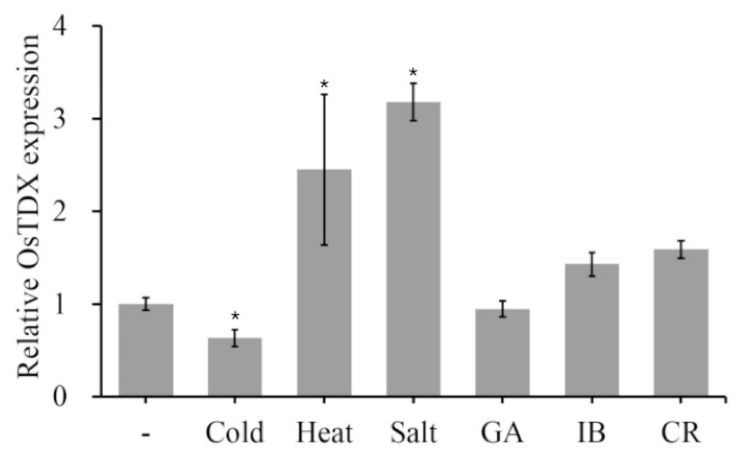
Analysis of *Oryza sativa* tetratricopeptide domain-containing thioredoxin (*OsTDX*) expression in response to abiotic stress. Rice seeds were irradiated with gamma ray (GA, 200 Gy), ion beam (IB, 40 Gy), and CR (cosmic rays). Ten-day-old rice seedlings were treated with heat (45 °C), cold (4 °C), and salt (200 mM NaCl). Error bars denote standard errors of biological replicates. Expression values of each gene are normalized against the expression of *OsActin.* Asterisks indicate statistical significance (*p* < 0.05, one-way ANOVA with a Tukey’s post hoc test) of differences between control and stress treatment.

**Figure 2 ijms-20-01413-f002:**
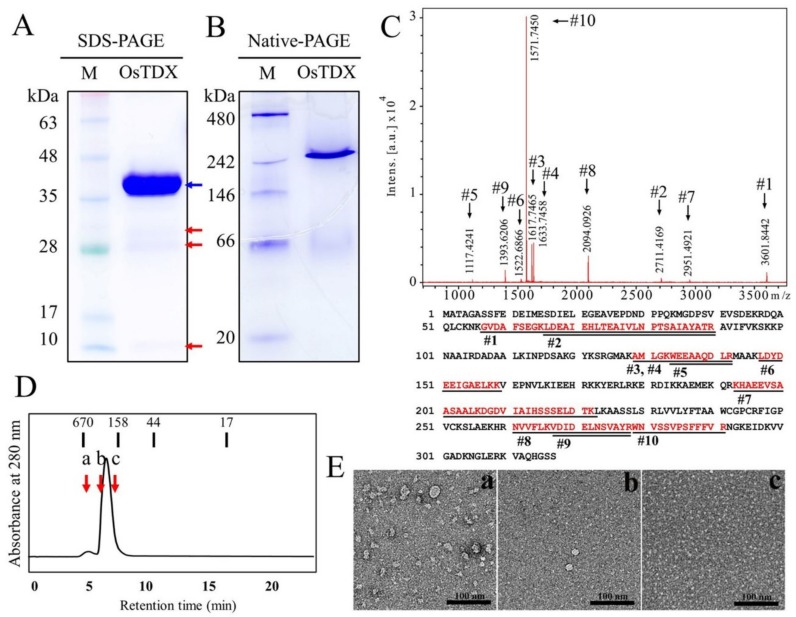
Purification and characterization of OsTDX protein. Bacterially expressed recombinant OsTDX was resolved by 12% SDS-PAGE (arrowheads indicated the fragmented OsTDX proteins) (**A**) and 10% native-PAGE (**B**). Coomassie-stained SDS-polyacrylamide gel (**A**) showing the purity of OsTDX after the final purification step (red arrow, truncated OsTDX protein). (**C**) The purified protein (blue arrow in Figure 1A) was identified by MALDI-TOF analysis as OsTDX (red). Each trypsin digested peptide fragment (defined as #) is indicated by the underlined amino acids. (**D**) Purified recombinant OsTDX protein was analyzed on SEC. The short vertical lines indicate standard markers that were calibrated with bovine thyroglobulin (670 kDa), bovine γ-globulin (158 kDa), chicken ovalbumin (44 kDa), and horse myoglobin (17 kDa). ((**E**) Oligomeric forms of OsTDX protein fractionated from SEC (a, b, and c red arrow in Figure 2D) were observed under TEM.

**Figure 3 ijms-20-01413-f003:**
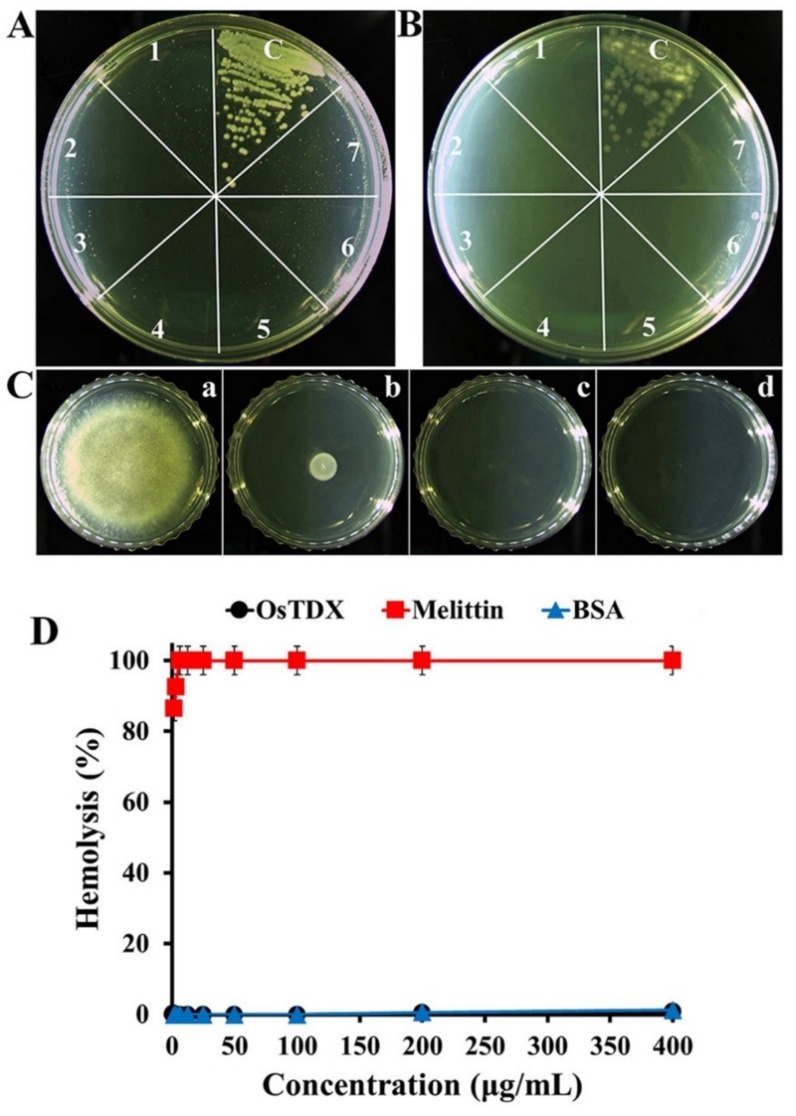
Antifungal activity of OsTDX protein against *Candida albicans* (**A**), *Candida krusei* (**B**), or *Colletotrichum gloeosporioides* (**C**) and cytotoxic effect of OsTDX, melittin, or BSA proteins against rat erythrocytes (**D**). (**A**,**B**) After 24-h incubation of fungal cells, in the presence of OsTDX, at the indicated concentrations, the cells were further incubated on agar medium for 24 h; c: control, 1: 200 µg/mL, 2: 100 µg/mL, 3: 50 µg/mL, 4: 25 µg/mL, 5: 12.5 µg/mL, 6: 6.25 µg/mL, 7: 3.13 µg/mL. (**C**) *C. gloeosporioides* conidia cells incubated with OsTDX protein for 24 h were dropped onto agar medium and further incubated for 72 h; a: control, b: 50 µg/mL, c: 100 µg/mL, d: 200 µg/mL. (**D**) After 2 h incubation of rat erythrocytes (8%, *v*/*v*) in the presence of proteins, the released hemoglobin was measured at 414 nm.

**Figure 4 ijms-20-01413-f004:**
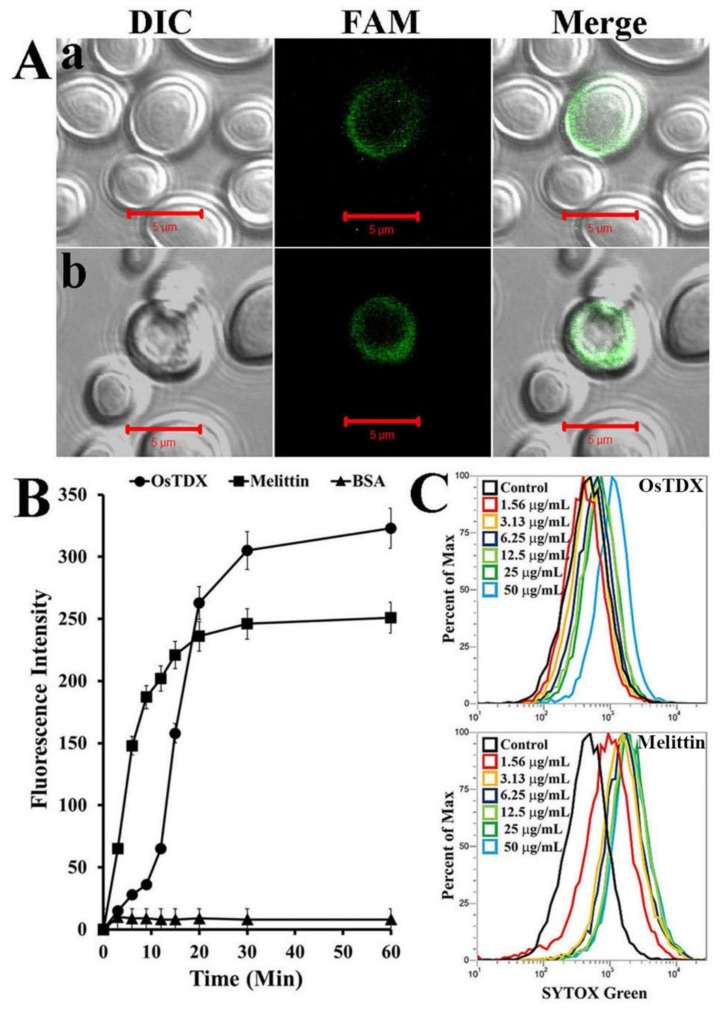
Cellular distribution and SYTOX-Green uptake of OsTDX in *Candida albicans* cells. (**A**) After 30 min incubation in the presence of FAM-labeled melittin (**a**) or OsTDX (**b**), *C. albicans* cells were washed with PBS buffer (pH 7.2) and observed under CLSM in differential interference contrast (DIC) and fluorescence conditions. (**B**) Time-dependent uptake of SYTOX-Green was measured using a fluorescence spectrophotometer after treatment of SYTOX-Green-pretreated *C. albicans* cells with melittin, OSTDX, and BSA. (**C**) After 30 min incubation in the presence of OsTDX or melittin at various concentrations, intracellular uptake of SYTOX-Green in *C. albicans* cells was measured by flow cytometry.

**Figure 5 ijms-20-01413-f005:**
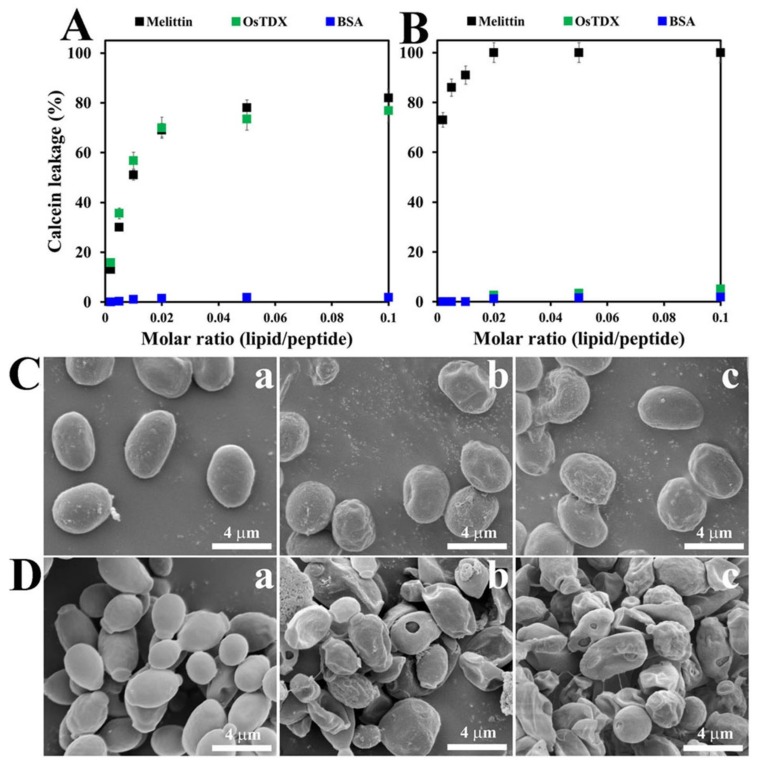
Membrane-permeabilizing activity of OsTDX in fungal cells. (**A**,**B**) Calcein-entrapped liposomes composed of PC/PE/PI/ergosterol (fungal membrane, 5:4:1:2, *w*/*w*/*w*/*w*, **A**) or PC/Ch/SM (mammalian membrane, 1:1:1, *w*/*w*/*w*, **B**) lipids were prepared, and liposome suspensions of 100 μM were incubated with indicated samples. The fluorescence of the released calcein was measured using a fluorescence spectrofluorometer, and 100% release was achieved using 0.03% Triton X-100. (**C**,**D**) Scanning electron micrographs of *Candida gloeosporioides* (**C**) or *Candida albicans* (**D**) cells treated with PBS buffer (**a**, control), OsTDX (**b**), or melittin (**c**) at minimum inhibitory concentration (MIC).

**Table 1 ijms-20-01413-t001:** Antifungal activity of OsTDX and melittin against various pathogenic fungi.

Fungi	MIC (µg/mL)		MIC (µM)	
	OsTDX	Melittin	OsTDX	Melittin
**Mold**				
*C. gloeosporioides*	100	25	2.86	8.78
*F. graminearum*	50	25	1.43	8.78
*F. oxysporum*	50	50	1.43	17.57
*F. solani*	100	50	2.86	17.57
*G. zeae*	100	100	2.86	35.14
**Yeast**				
*C. albicans*	1.56	6.25	0.04	2.2
*C. krusei*	1.56	6.25	0.04	2.2
*C. parapsilosis*	6.25	12.5	0.18	4.39
*T. beigelii*	6.25	12.5	0.18	4.39

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
