# Peer review of "Functional Mechanisms Underlying the Antimicrobial Activity of the Oryza sativa Trx-like Protein"

_ijms, 2019, doi:10.3390/ijms20061413_

Round 1

Reviewer 1 Report

The manuscript by Park et al. report the isolation, purification, characterization of Oryza sativa Trx-Like protein which demonstrated antimicrobial activity in the selected strains of various fungus. The mechanism of antimicrobial activity was investigated using appropriate control. The cellular uptake and membrane permeability were performed.  The study design was well planned. The manuscript is well written and suitable for the International journal of molecular sciences. However, some minor revision need to make in light of the following comments.,

1.      The author should add information for the source and origin of Oryza Sativa L. seeds used in the experiment. Do all the experiments were performed from a single batch of protein or isolated differently?

2.      Please add more detail analysis of different peaks in the MALDI spectra to the result of the discussion section (line 85).

3.      What were the control drugs used in the antifungal assay? Author should report them. Standard deviation or errors need to mention in these data.

4.      Fig 3D, please use a different color for compound so more visible for the reader.

Author Response

The manuscript by Park et al. report the isolation, purification, characterization of Oryza sativa Trx-Like protein which demonstrated antimicrobial activity in the selected strains of various fungus. The mechanism of antimicrobial activity was investigated using appropriate control. The cellular uptake and membrane permeability were performed.  The study design was well planned. The manuscript is well written and suitable for the International journal of molecular sciences. However, some minor revision need to make in light of the following comments,

Q1).      The author should add information for the source and origin of Oryza Sativa L. seeds used in the experiment. Do all the experiments were performed from a single batch of protein or isolated differently?

A1) According to reviewer’s comment, this information was added to the “Materials and Methods” section in the revised manuscript (line 207). All experiments were performed at least three times by using the OsTDX protein from single batch purification and the results were same.

Q2).      Please add more detail analysis of different peaks in the MALDI spectra to the result of the discussion section (line 85).

A2) According to reviewer’s comment, we added more detailed explanation for the MALDI spectra data in line 87-88. And we also provide more detail experimental methods for the MALDI-TOF analysis in "materials and methods" section (line 253-255).

Q3)      What were the control drugs used in the antifungal assay? Author should report them. Standard deviation or errors need to mention in these data.

A3) Melittin peptide, that is a small linear cytolytic peptide composed of 26 amino acid residues, was used as a control drug because OsTDX is a protein with antifungal activity and we want to compare antifungal activity of well-known material composed of amino acids. “melittin, a cytotoxic peptide used as a control drug” was already described in Line 111-112.

In antifungal assay, samples were serially two-folded diluted in 96-well plate and the MICs were defined as the lowest concentration of samples that completely inhibited the visible growth. MICs cannot express in standard deviation or errors because growth inhibitory rates were not calculated to percentage by measurement of absorbance or staining methods, and each MICs were same against fungal strains in triplicated experiments.

Q4.      Fig 3D, please use a different color for compound so more visible for the reader.

A4) According to reviewer’s comment, we represented figure 3D to visible color.

Reviewer 2 Report

The manuscript by Park, Kim et al. describes identification, isolation and partial characterization of a thioredoxin-like protein that could play a role in plant-pathogen interactions. Authors employed multiple complementary techniques and concluded that this protein is an antifungal agent. However, the conclusions are based on an assumption, that the recombinant protein is pure. The presented PAGE analyses clearly indicate co-purified proteins, and this seriously undermines the validity of presented results, especially at higher concentrations employed in the antifungal assay. I believe that authors should try to improve the protein purity (e.g. SCX chromatography) or prove that contaminants (likely of bacterial origin) do not show similar antifungal effects. Authors should also address the possibility that TDX at higher concentration could be toxic to plant itself. Do plants survive its presence in growth media? It would be also interesting to determine/discuss protein localization - is it an extracellular protein?

Minor issues:

Abstract

Please, rephrase the sentence implying that environmental conditions are an insult.

Results

Authors indicate that they have profiled redox-related genes, but the presented results are only for TDX. It is not clear what criteria were employed to identify "rice redox genes", and why only a single protein/gene was selected. 

 Figure 1 is missing statistical analysis - ANOVA and posthoc analysis should be provided to support the results.

The combination of mass, charge and the 3D structure drives the protein separation in a native PAGE. If the presented figure were to be a native PAGE, it would not be possible to determine the molecular mass without a Ferguson plot.

Figure 2D is poorly described in the manuscript 

Methods

Authors should include a description/reference for all experiments. Some experiments are not described at all (e.g. native gell separation, MALDI analysis), and some descriptions are only rudimentary (e.g. parameters for SEC separation). 

Author Response

The manuscript by Park, Kim et al. describes identification, isolation and partial characterization of a thioredoxin-like protein that could play a role in plant-pathogen interactions. Authors employed multiple complementary techniques and concluded that this protein is an antifungal agent. However, the conclusions are based on an assumption, that the recombinant protein is pure. The presented PAGE analyses clearly indicate co-purified proteins, and this seriously undermines the validity of presented results, especially at higher concentrations employed in the antifungal assay. I believe that authors should try to improve the protein purity (e.g. SCX chromatography) or prove that contaminants (likely of bacterial origin) do not show similar antifungal effects.

A) Before submission, we already confirmed that the all fragments on SDS-PAGE gel were OsTDX proteins, resulting from the MALDI-TOF analysis (Figure 2A: SDS-PAGE). However, as a reviewer suggested, we added more clear SDS-PAGE data in our revised manuscript. All activities of new protein (revised data) were exactly same with old protein (original SDS-PAGE data).

Authors should also address the possibility that TDX at higher concentration could be toxic to plant itself. Do plants survive its presence in growth media? It would be also interesting to determine/discuss protein localization - is it an extracellular protein?

A) We are using the Arabidopsis plants overexpressing the OsTDX protein in our other project. The OsTDX overexpressed Arabidopsis plants show no difference with wild type plant and/or are more healthy than wild type plants. So, we do not need to add the toxicity in OsTDX overexpressed plants in the revised manuscript. A result about subcellular localization of OsTDX has been preparing for another publication.

Minor issues:

Abstract

Please, rephrase the sentence implying that environmental conditions are an insult.

A) According to reviewer’s comment, we revised the sentence in the “Abstract” section (line 16-18).

Results

Authors indicate that they have profiled redox-related genes, but the presented results are only for TDX. It is not clear what criteria were employed to identify "rice redox genes", and why only a single protein/gene was selected. 

A) We investigated the effect of environmental stresses on the expression of rice redox genes including several rice genes of Trx fold containing proteins. Especially, the transcription of a rice OsTDX was significantly changed in response to several stresses compared with other rice redox genes and OsTDX protein showed strong antimicrobial activity. As RT-PCR data of other rice redox genes have already used for publication process of another paper, we cannot use it unfortunately.

 Figure 1 is missing statistical analysis - ANOVA and post hoc analysis should be provided to support the results.

A) As suggested, the result have been treated by statistical analysis using one-way ANOVA analysis with a Tukey’s post hoc test, and modified Figure 1 and the legend on revised manuscript as follows:

In the Figure legend of Figure 1                       

“Asterisks indicate statistical significance (P < 0.05, one-way ANOVA with a Tukey’s post hoc test) of differences between control and stress treatment.”

The combination of mass, charge and the 3D structure drives the protein separation in a native PAGE. If the presented figure were to be a native PAGE, it would not be possible to determine the molecular mass without a Ferguson plot.

A) The purpose of native PAGE, size exclusion chromatography (SEC) and TEM analysis was only for showing natural structures of OsTDX protein like monomer, homo-oligomer and so on. We revised the sentence (line 87-88).

Figure 2D is poorly described in the manuscript 

A) According to reviewer’s comment, we added the sentence to describe SEC analysis in line 91-93.

Methods

Authors should include a description/reference for all experiments. Some experiments are not described at all (e.g. native gel separation, MALDI analysis), and some descriptions are only rudimentary (e.g. parameters for SEC separation). 

A) According to reviewer’s comment, we revised “Materials and Methods” section and add several references (ref 26-30).

Reviewer 3 Report

The manuscript by Seong-Cheol Park etc. evaluated the antimicrobial activity and mechanism of OsTDX protein. This topic is interesting because the studies provide hints in that the plant redox proteins induced by exotic stress are functionally related to the defense responses. A serial of experiments was conducted to prove the antifungal activity of OsTDX protein. The mechanism of function was supported orthogonally by several methods, which unambiguously indicated that the OsTDX protein kills fungal pathogens via destabilizing and disrupting effects on fungal membranes.  

The reviewer agrees that the manuscript is suitable for the scope of Int. J. Mol. Sci., and the report was generally well written, hence suggests acceptance of the manuscript with minor revision. 

1.       The introduction should be improved, it is awkward to proceed to paragraph 3 from paragraph 2. The reviewer suggests the authors reword or add more words to better describe the significance of this work in the introduction.

2.       The MALDI data in Figure 2C was not clearly readable, please revise on this.

Author Response

Q1. The introduction should be improved, it is awkward to proceed to paragraph 3 from paragraph 2. The reviewer suggests the authors reword or add more words to better describe the significance of this work in the introduction.

A1) According to reviewer’s comment, we revised the paragraph in “Introduction” section.

Q2. The MALDI data in Figure 2C was not clearly readable, please revise on this.

A2) According to reviewer’s comment, we revised Figure 2C data.

Round 2

Reviewer 2 Report

The revision of the manuscript does not address all my concerns:

R1: Before submission, we already confirmed that the all fragments on SDS-PAGE gel were OsTDX proteins, resulting from the MALDI-TOF analysis (Figure 2A: SDS-PAGE). However, as a reviewer suggested, we added more clear SDS-PAGE data in our revised manuscript. All activities of new protein (revised data) were exactly same with old protein (original SDS-PAGE data).

- The PAGE image quality is quite low with bluish background. It is quite strange that the "problematic" part around 30 kD is blank white. If authors believe that lower mass bands are part of recombinant OsTDX (degradation products?), it would be more productive to preserve the original image and include above mentioned reports from MALDI analysis. Your claim that all bands were analyzed is not supported by the presented results. 

R2: We investigated the effect of environmental stresses on the expression of rice redox genes including several rice genes of Trx fold containing proteins. Especially, the transcription of a rice OsTDX was significantly changed in response to several stresses compared with other rice redox genes and OsTDX protein showed strong antimicrobial activity. As RT-PCR data of other rice redox genes have already used for publication process of another paper, we cannot use it unfortunately.

- Your manuscript inludes e.g. this sentece: "In this study, we investigated the transcription levels of rice genes that encode ROS regulators in response to environmental stresses" - this is clearly not true. If you want to present only a single gene, do explain the selection and don't indicate that you have results of something that you don't have/don't want to include in the manuscript.  

R3: The purpose of native PAGE, size exclusion chromatography (SEC) and TEM analysis was only for showing natural structures of OsTDX protein like monomer, homo-oligomer and so on. We revised the sentence (line 87-88).

- Your motivation does not change the fact that this is wrong. You can't estimate MW from native PAGE like this. 

R4: According to reviewer’s comment, we revised “Materials and Methods” section and add several references (ref 26-30).

- Chapter title is missleading (Cloning of the OsTDX gene and protein expression in E. coli) and referenced manuscript don't provide sufficient information to reproduce your results (e.g. there is nothign about MALDI-MS instrument type, settings, etc.)

Author Response

R1: The PAGE image quality is quite low with bluish background. It is quite strange that the "problematic" part around 30 kD is blank white. If authors believe that lower mass bands are part of recombinant OsTDX (degradation products?), it would be more productive to preserve the original image and include above mentioned reports from MALDI analysis. Your claim that all bands were analyzed is not supported by the presented results.

=> According to reviewer’s comment, we changed Fig. 2A with an original SDS-PAGE gel image and revised sentences in “Results” section. We added the MALDI-TOF results of OsTDX partial fragments (revised Fig. 1A) in “Results” and “Supplementary data” section (Fig. S2) (line 87-91).

R2: Your manuscript includes e.g. this sentece: "In this study, we investigated the transcription levels of rice genes that encode ROS regulators in response to environmental stresses" - this is clearly not true. If you want to present only a single gene, do explain the selection and don't indicate that you have results of something that you don't have/don't want to include in the manuscript.

=> We honestly performed quantitative RT-PCR analysis with several rice redox genes to investigate the effect of various stresses. As OsTDX gene was dramatically induced by various stresses, we chose the OsTDX protein for this study. According to reviewer’s comment, we added some of qRT-PCR results in supplementary data (Fig. S1) (line 72-73). As we already mentioned that qRT-PCR data of the other rice redox genes have already used for another paper, we could use a part of qRT-PCR sets for this paper unfortunately.

R3: Your motivation does not change the fact that this is wrong. You can't estimate MW from native PAGE like this.

=> We agreed reviewer’s comment that native-PAGE data can not be used to estimate MW. According to reviewer’s comment, we revised all sentences about MW estimation one by one.

R4: Chapter title is missleading (Cloning of the OsTDX gene and protein expression in E. coli) and referenced manuscript don't provide sufficient information to reproduce your results (e.g. there is nothing about MALDI-MS instrument type, settings, etc.)

=> According to reviewer’s comment, we divided section 3.4. to “3.4. Cloning of the OsTDX gene and protein expression in E. coli” and “3.5. Purification and structural characterization of OsTDX protein” in revised “Materials and Methods” (line 252-262). We also added sufficient information about MALDI-MS instrument type and settings in revised “Materials and Methods” (line 271-274).

Round 3

Reviewer 2 Report

All my major issues were addressed and I don't believe that any further comments could improve this manuscript.